# Viscoelastic Properties of Water-Absorbed Poly(methyl methacrylate) Doped with Lithium Salts with Various Anions

**DOI:** 10.3390/molecules27207114

**Published:** 2022-10-21

**Authors:** Asae Ito, Arisa Shin, Koh-hei Nitta

**Affiliations:** Polymer Physics Laboratory, Institute of Science and Engineering, Kakuma Campus, Kanazawa University, Kanazawa 920-1192, Ishikawa, Japan

**Keywords:** dynamic mechanical properties, lithium salts, pinning effect, poly (methyl methacrylate)

## Abstract

We investigated the effects of water absorption on the dynamic mechanical properties of poly(methyl methacrylate) doped with various generic lithium salts, such as lithium perchlorate trihydrate (LiClO_4_), lithium trifluoromethanesulfonate (LiCF_3_SO_3_), lithium nonafluorobutanesulfonate (LiC_4_F_9_SO_3_), and lithium bis(trifluoromethanesulfonyl)imide (LiN(CF_3_SO_2_)). The rates of weight change during water absorption of lithium salt-doped samples were higher in the following order: LiClO_4_, LiCF_3_SO_3_, LiC_4_F_9_SO_3_, and LiN(CF_3_SO_2_). Interestingly, the aforementioned order was the same as the order of the terminal relaxation times in the flow region of the viscoelastic measurement in the melting-state. This implies that the water absorption of the salt-doped PMMA occurs due to the factors that affect the pinning of the PMMA molecular chains in the places.

## 1. Introduction

It is common knowledge that generic lithium salts, typically used as polymer electrolytes, exhibit a well-developed aggregate structure in a polymer matrix [1,2]. In recent years, examples of the use of glassy polymers as electrolytes have been reported: a transparent PMMA-based gel electrolyte [3], polymer-based electrolytes using PMMA-based copolymers introducing LiCF_3_SO_3_ (lithium trifluoromethanesulfonate; LFMS) with high ionic conductivity [4], PEO-PMMA and alumina-based electrolyte doped with lithium (istrifluoromethanesulfonimide lithium) (LiTFSI) with a high electrical conductivity [5].

In a previous study, we demonstrated that the well-aggregated salts interact strongly with the polar groups of PMMA in compression-molded sheets, increasing *T*_g_ [6,7,8], decreasing birefringences [9], and enhancing brittleness [10]. According to the rheological spectra at compression-molding temperature, the doping with salts was found to prolong the relaxation times in the glassy and flow regions at the compression-molding temperature. These results imply that both the segmental and macro-Brownian motions of PMMA chains were suppressed in the matrix during the comp-molding process [10]. Moreover, we found that the mechanical properties of the compression-molded PMMA sheets after rapid cooling are directly influenced by the relaxation times in the flow regions at the compression-molded temperature [10]. These findings resulted in the conclusion that the brittleness of the PMMA sheets is dominated by the molecular morphology in the glassy state reflecting the molecular mobility under the compression-molded process.

Moisture absorbability of PMMA makes it difficult to control the mechanical properties, such as toughness and brittleness, because of its unstable mechanical characteristics based on the circumstances. Ishiyama et al. [11] conducted tensile tests for PMMA at three different elongation speeds under humidity conditions, and they found that the Young’s modulus of PMMA increases linearly with decreasing humidity. 

Doping of the salts enables us to control the water absorption concentration in the PMMA matrix using the high-water absorbency of generic lithium salts. We have previously reported that water-absorbed PMMA-salt samples revealed a novel peak shoulder at the *T*_g_ relaxation mode in the viscoelastic spectra in a solid state [6]. Thus, this study focuses on the effects of absorbed water on the rheological and dynamic mechanical properties of PMMA doped with lithium salts.

Furthermore, “water-in-salt” electrolyte has received a lot of attention from the point of view of thermal stability in recent years [12,13,14]. Given the circumstances, our study also provides a simple method for enhancing PMMA’s water retention capacity using lithium salts, thus expanding future materials design possibilities.

## 2. Results and Discussion

Figure 1a–d compares the dynamic mechanical spectra of the salt-doped PMMA solids for dried and moisten sheets. The dynamic mechanical spectra of PMMA are shown in Appendix A. Table 1 lists the water absorption contents of these sample sheets. We have previously reported that PMMA and salt-doped PMMA samples plasticize under constant humidity. In PMMA, there are typically two relaxation *E″* peaks—*α* (around at 100–120 °C) and *β* (around at −50–100 °C). The broad peak around 100–120 °C appeared due to the overlapping of dual *a*-relaxation (glass transition) peaks above 100 °C; the peak in the lower temperatures is attributed to dried PMMA domains, whereas the one lower is attributed to water-absorbed PMMA domains due to the plasticization by water. It has been reported that a novel shoulder peak (*β*′), which partially overlaps the *β* relaxation, appeared due to moisture absorption at the lower temperature side of the *β* relaxation peak [15,16]. This phenomenon also depends on the molecular weight of PMMA, according to Shen et al. [17]. Figure 1 shows that the water absorption also increased *β* and *β*′ relaxation, which is consistent with Ceccorulli’s report [16]. They also reported that the increase in *β*′ is due to some association of water–water molecules partially interacting with the ester units of PMMA [16] based on “complex relaxation” [18]. 

Figure 2a–d compare the master curves at 200 °C in the melting-state viscoelastic measurements between pristine and salt-doped PMMA. Notably, the reference temperature (200 °C) is the compression-molding temperature for preparing the sheets for tensile tests. The dried samples were used for the rheology measurements.

A significant increase in the average relaxation times in the glassy region was observed for three salt-doped samples (PMMA/LiClO_4_, PMMALiCF_3_SO_3_, and PMMA/LiC_4_F_9_SO_3_), and the terminal relaxation time zone was prolonged. Alternatively, the master curve in PMMA/LiN(CF_3_SO_2_)_2_ almost overlapped with the master curve of the pristine PMMA, and the terminal relaxation was almost the same as that of pristine PMMA, but the tan *δ* peak was higher than that of pristine PMMA. 

Table 2 adds the specific ratio of the mean relaxation time in the glassy region (<*τ*_G_>/<*τ*_G0_>), the specific entanglement density (<*ν*_e_>/<*ν*_e0_>) and the specific ratio of mean relaxation times in the flow region (<*τ*_F_>/<*τ*_F0_>), where the suffix 0 in these specific ratios denotes those three parameters of the pristine PMMA. The analytical details for estimating these three parameters are presented in our previous paper [7,8,10]. Here, the overall master curves, including the relative relaxation times in the glass transition and the flow regions, shift to longer time regions due to the addition of these salts. These longer time shifts are responsible for ion–dipole interactions between carbonyl groups of PMMA and ions, demonstrating the pinning effects on PMMA chains, as shown in previous studies.

Here, these parameters are independent of the anion radius or ovality of the anions, implying that these salts aggregate in PMMA, which is also consistent with our previous report [7,8].

Figure 3 shows the specific relaxation times in the terminal zone of dried samples plotted against the water absorption contents. The terminal relaxation region elongates to longer periods as the water absorption progresses. The aggregated salts with the pinning effects of the absorbed water and the free salts in the PMMA matrix are isolated in the PMMA matrix because the pinning effects of PMMA chains appear in the terminal zone for Li-salt-doped PMMA. Further, the salt-doping process makes it possible to retain the water content in the PMMA matrix and to control the rheological properties via the pinning effects. It is considered that the stronger pinning effects on the molecular chains imply that the aggregation of the present salts is weaker than that of the other salts, making them absorb water easily.

We performed the tensile tests of PMMA doped with salts before and after moisture absorption (see Appendix A). Table 3 summarizes the averaged value of toughness, which is the area under stress–strain curve up to break, and the terminal relaxation times. The sheets doped with the salts with higher pinning effects, i.e., LiClO_4_ and LiCF_3_SO_3_, in the dried sample had higher toughness due to moisture absorption. Alternatively, the sheets doped with salts with lower pinning effects, i.e., C_4_F_9_SO_3_ and LiN(CF_3_SO_2_)_2_, had lower toughness values after moisture absorption. This is because the salts with stronger pinning effects possess higher water-absorbability around the pinning position. We previously showed that salt-doped PMMA becomes brittle [7,8,10]. However, the embrittlement is suppressed in samples where water is more easily adsorbed to the pinning positions of the salts since they are plasticized by moisture absorption. These results imply that the mechanical properties are dominated by the competition among the three-body interactions of PMMA, salt, and water.

## 3. Experimental Procedure

### 3.1. Materials and Sample Preparation

PMMA pellets (*M*_w_ = 1.0 × 10^5^ and *M*_w_/*M*_w_ = 1.9) calibrated using a PMMA standard were used in this study. The four types of lithium salts used in this study include the following: lithium perchlorate trihydrate (LiClO_4_, Nacalai Tesque, inc., Kyoto, Japan), lithium trifluoromethanesulfonate (LiCF_3_SO_3_, purity ≥ 98.0%; Tokyo Chemical Industry Co. Ltd. (*TCI*), Tokyo, Japan), lithium nonafluorobutanesulfonate (LiC_4_F_9_SO_3_, purity ≥ 95.0%; Tokyo Chemical Industry Co. Ltd. (*TCI*)), and lithium bis(trifluoromethanesulfonyl)imide (LiN(CF_3_SO_2_), purity ≥ 98.0%; Tokyo Chemical Industry Co. Ltd. (*TCI*)), without any additional purification. Figure 4 shows their chemical structures. The PMMA sheets doped with these salts were prepared for solution casting using a mixture of dichloromethane and methanol at a weight ratio of 9:1 for 1 h. The salt concentrations for each sheet were fixed at molar ratios of 0.07 in PMMA for each blend. The salt-doped PMMA sheets were dried at 135 °C for 30 h to evaporate the residual solvents after being dried in a draft chamber for 1 day. Approximately 200-µm-thick sample sheets with salts were obtained by compression-molding at 200 °C and 20 MPa for 5 min after preheating at 200 °C for 5 min and rapid cooling at 25 °C for 5 min.

### 3.2. Characterization

The dried sample sheets were stored in a desiccator with 20% relative humidity (RH) until just before each measurement.

The moisture sheets were stored under the conditions of the temperature of 23 °C and the relative humidity of 70% RH. The amount of absorbing water in these PMMA sheets was estimated from the weight changes using an electronic balance (ASR224/E, Kanazawa, Japan). Figure 5 shows the weight changes during water absorption [6]. The sample weight increased as time increased and reached an equilibrium value after 30 min for all salt-doped samples. Table 1 lists the equilibrium absorbed water content. The salt-dopped PMMA sheets absorbed more water than pristine PMMA, in which the water absorption was in the following order: PMMA/LiClO_4_ > PMMA/LiCF_3_SO_3_ > PMMA/LiC_4_F_9_SO_3_ > PMMA/LiN(CF_3_SO_2_)_2_, at the salt concentration of 0.07 molar ratio. Moisture samples in the equilibrium state were used for the measurements. The sample codes in Table 1 indicate the dried and the equilibrium water-absorbed (w) ones.

### 3.3. Measurements

A viscoelastic spectrometer (DVE-V4, UBM Co. Ltd., Kyoto, Japan) was used to conduct solid state dynamic mechanical analysis. The temperature dependence of the dynamic mechanical properties was obtained in the temperature range of −150–200 °C at 2 °C/min and 10 Hz. The distance between the chucking apparatus was 20 mm.

The melting-state viscoelastic measurement was conducted using a rotational rheometer (Discovery HR-2, TA Instruments, New Castle, DE, USA) and a parallel plate with a diameter of 8 mm under nitrogen flow. The initial gap distance was 1000 µm, the angular frequency was 0.1–100 rad/s, and the temperature range was 120–240 °C (at 10 °C increments).

The tensile test was conducted at an elongation speed of 10 mm/min using a tensile testing machine (TC 05–010, Abe Seisakusho, Kanazawa, Japan). Thin rectangular specimens with gauge size 5 mm × 10 mm were cut from the sample sheets using an ultrasonic cutter.

## 4. Conclusions

This study showed that the PMMA samples doped with lithium salts with a high pinning effect exhibited the highest levels of water absorption. This implies that the salts with stronger interaction with PMMA and less tendency to aggregate in PMMA have superior water retention in PMMA. Further, the salts with stronger pinning effects can enhance PMMA’s fracture energy after water absorption compared with other salts. Therefore, the results are also interesting for the industry because they demonstrate that fracture toughness in glassy polymers, such as PMMA and polystyrene, can be improved by adequate water retention.

## Figures and Tables

**Figure 1 molecules-27-07114-f001:**
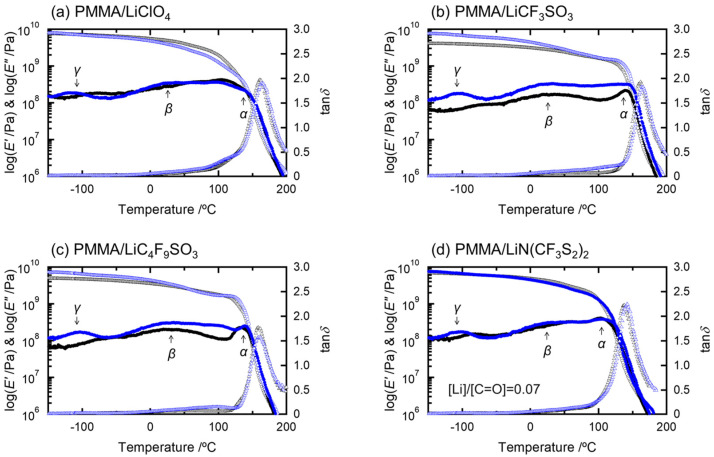
Temperature dependence of dynamic mechanical properties of dried (black) and water-absorbed (blue) samples: PMMA doped with (**a**) LiClO_4_, (**b**) LiCF_3_SO_3_, (**c**) LiC_4_F_9_SO_3_, and (**d**) LiN(CF_3_SO_2_)_2_, with a 0.07-molar ratio salt concentration. *α* relaxation around at 100–120 °C is ascribed to the glass transition; *β* relaxation around at −50–100 °C is to the relaxation of side groups; *γ* relaxation around at −150–−50 °C is to the relaxation of local relaxation mode.

**Figure 2 molecules-27-07114-f002:**
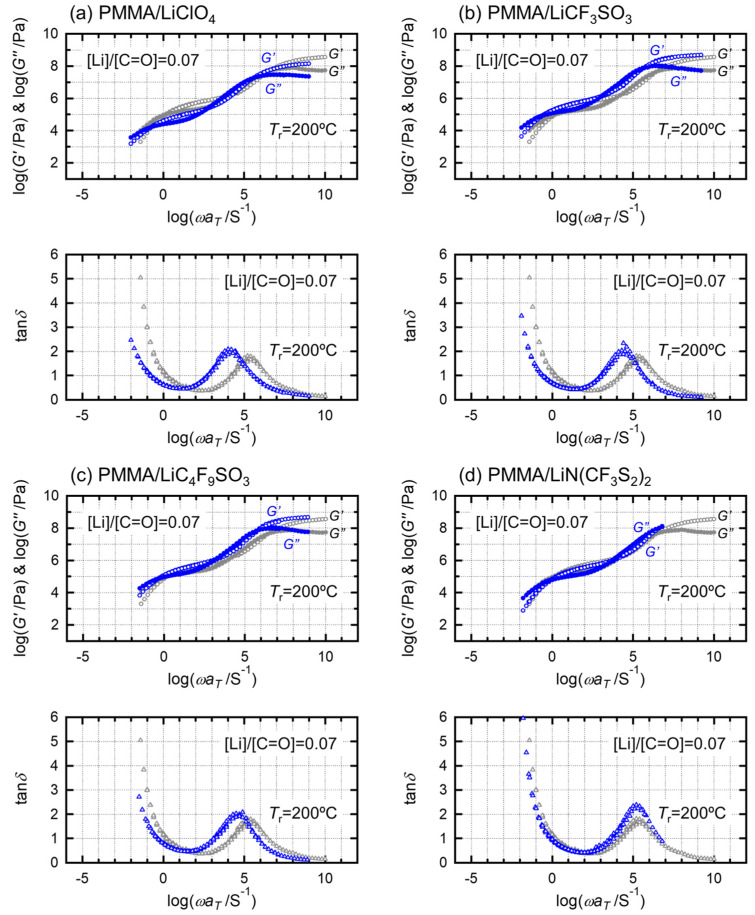
Dynamic viscoelastic spectra in the melts. PMMA (gray) and PMMA doped with the different lithium salts (blue): (**a**) LiClO_4_, (**b**) LiCF_3_SO_3_, (**c**) LiC_4_F_9_SO_3_, and (**d**) LiN(CF_3_SO_2_)_2_, with a 0.07 molar ratio salt concentration. The open circle denotes *G′*; the closed one denotes *G″*; the open triangle denotes tan *δ*.

**Figure 3 molecules-27-07114-f003:**
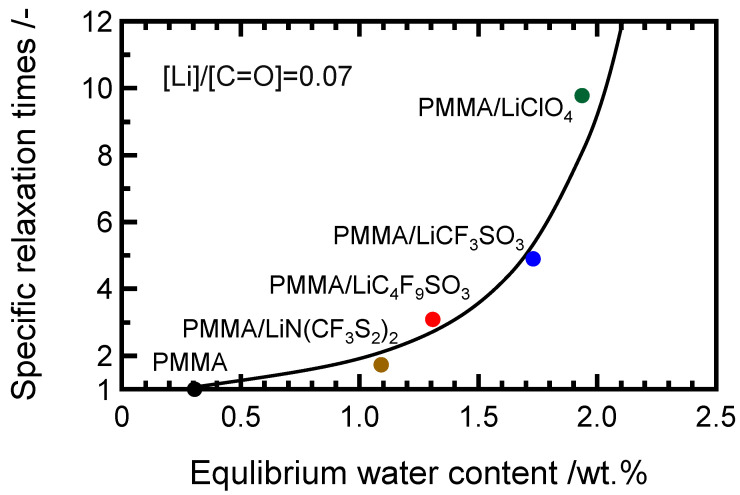
Equilibrium water content plotted against relative relaxation time in the terminal zone of PMMA doped with LiClO_4_ (green), LiCF_3_SO_3_ (blue), LiC_4_F_9_SO_3_ (red), and LiN(CF_3_SO_2_)_2_ (brown) with a 0.07 molar ratio salt concentration stored at 23 °C and 70% RH. The black symbol shows the value of PMMA.

**Figure 4 molecules-27-07114-f004:**
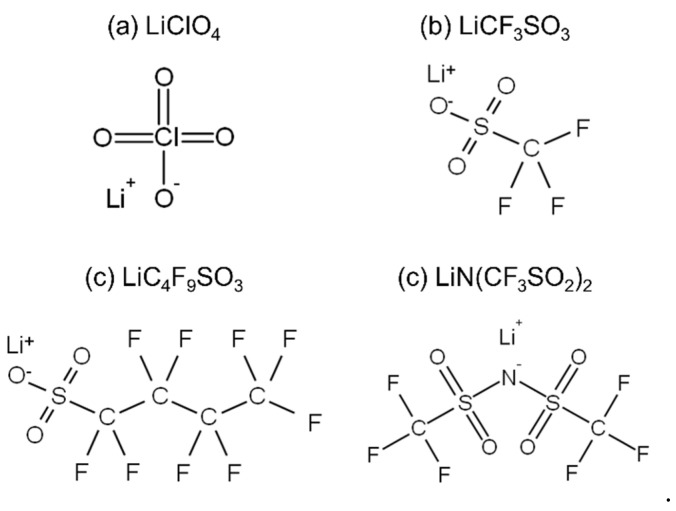
Lithium salts used in this study: (**a**) LiClO_4_, (**b**) LiCF_3_SO_3_, (**c**) LiC_4_F_9_SO_3_, and (**d**) LiN(CF_3_SO_2_)_2_.

**Figure 5 molecules-27-07114-f005:**
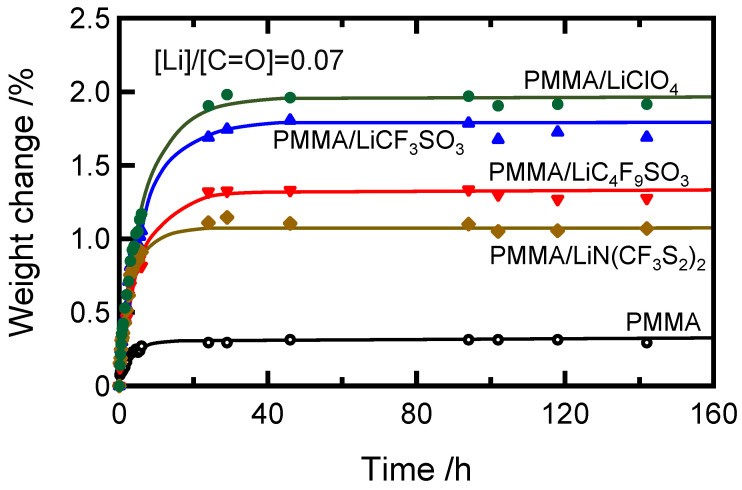
Weight changes in PMMA doped with LiClO_4_, LiCF_3_SO_3_, LiC_4_F_9_SO_3_, and LiN(CF_3_SO_2_) at 23 °C and 70% RH.

**Table 1 molecules-27-07114-t001:** The sample code, salt concentrations (weight and molar concentrations), and moisture concentrations of the dried and the equilibrium saturated moisture contents of the PMMA sheets doped with salts.

Sample Code	Condition	Weight Percent of Salt/wt.%	Molar Ratio of Salt/mol mol^−1^	Absorbed Water Content/wt.%
PMMA	Dried	0	0	0
	Water absorbed			0.31
PMMA/LiClO_4_	Dried	7	0.07	0
	Water absorbed			0.19
PMMALiCF_3_SO_3_	Dried	10	0.07	0
	Water absorbed			1.7
PMMA/LiC_4_F_9_SO_3_	Dried	18	0.07	0
	Water absorbed			1.3
PMMA/LiN(CF_3_SO_2_)_2_	Dried	17	0.07	0
	Water absorbed			1.1

**Table 2 molecules-27-07114-t002:** The specific ratio of the mean relaxation times in glassy and flow regions and the specific entanglement densities of the dried salt-doped samples. The suffix 0 indicates the pristine-PMMA.

Sample Code	<*τ*_G_>/<*τ*_G0_>	<*ν*_e_>/<*ν*_e0_>	<*τ*_F_>/<*τ*_F0_>
PMMA	1	1	1
PMMA/LiClO_4_	8.9	0.14	9.8
PMMALiCF_3_SO_3_	6.3	0.83	4.9
PMMA/LiC_4_F_9_SO_3_	3.2	0.81	3.1
PMMA/LiN(CF_3_SO_2_)_2_	0.63	0.72	1.7

**Table 3 molecules-27-07114-t003:** Toughness values of dried samples and water-absorbed samples PMMA and PMMA doped with LFMS, LFBS, LiClO_4_, and LiN(CF_3_SO_2_)_2_ at a molar concentration of 0.07.

Sample Code	Condition	Toughness/MJm^−3^
PMMA	Dried	7.4
	Water absorbed	8.3
PMMA/LiClO_4_	Dried	4.2
	Water absorbed	4.7
PMMALiCF_3_SO_3_	Dried	3.7
	Water absorbed	5.5
PMMA/LiC_4_F_9_SO_3_	Dried	3.8
	Water absorbed	2.2
PMMA/LiN(CF_3_SO_2_)_2_	Dried	4.9
	Water absorbed	4.0

## Data Availability

Not applicable.

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
