# Peer review of "Viscoelastic Properties of Water-Absorbed Poly(methyl methacrylate) Doped with Lithium Salts with Various Anions"

_molecules, 2022, doi:10.3390/molecules27207114_

Round 1

Reviewer 1 Report

To authors

This study reported that the effect of water absorption on dynamic mechanical properties of poly(methyl methacrylate) (PMMA) doped with various lithium salts. Since PMMA is one of transparent plastics with higher hardness, the obtained results in this study are useful to develop the material design, which are applied in practical use. The experimental data were carefully analyzed and discussed in multiple directions. Especially, the universal relation between the water absorption ability and melt relaxation time is important because the two parameters are not directly associated. From these viewpoints, the reviewer recommends to accept the manuscript in this journal. However, some parts should be revised to address reviewer’s comments.

1. Page 3, line 80: In the first sentence. What is “Table 25. C and 70% RH”? It may be the experimental condition for water absorption.

2. Page 3, line 88: The end of paragraph seems to be connected with Figure 2. After the end of paragraph, line fees should be inserted.

3. Page 4, line 110: In the sentence, “dried and moisture sheets” should be modified as “dried and moisten sheets”.

4. Figures 3 and 4: In the caption, difference in the colors for data plots should be indicated. Both the figures were plotted with black and blue colors in PDF files.

5. Page 5, line 137: The authors mentioned as “A significant increase in Tg was observed for three salt-doped samples (PMMA/LiClO4, PMMALiCF3SO3, and PMMA/LiC4F9SO3), …” from Figure 4. However, the figure represented the frequency dependence of complex modulus and loss tangent. How was the representation obtained from the experimental data? Please add some explanations.

6. Page 6, line 149: “This longer time shifts are…” should be “These longer time shifts are…”

7. Page 6, Figure 5: In the vertical axis, “Specific relaxation time / s” should be “Specific relaxation time” without unit. Furthermore, was the solid curve theoretically obtained? The reviewer wondered why the line did not started from the plot for pure PMMA. 

Author Response

Dear reviewer 1

Thank you for your comments. I send you a reply.

Thank you for your kind treatments. I have apologized my delay response. I received the review result on 20th September because of unexpected system errors.

Reviewer 2 Report

The manuscript presents really interesting data on the mechanical properties of PMMA + water + lithium salts. Besides the good experimental data, the manuscript should be revised before publication. The English language and style must be improved. In some parts, the text sounds colloquial. The methodology is poorly described and the results are not properly discussed. I suggest the authors make a literature review to better explain the obtained results.

The introduction in general does not provide a good background for the reader. Authors cite: “PMMA has been widely used for medical applications”, “PMMA-based composites have been investigated for use in dentistry using additives such as methacryloxyethyl trimellitic anhydride and hydroxyapatite”. It is important to address the perspectives found by the previous studies on PMMA + water +lithium salts: what are the problems of using this mixture? Why is it still necessary to enhance the properties of PMMA?

Lines 24 and 38 - First and second paragraphs start with: “it is common knowledge”. Remove this expression.

Methods

Line 61 - Specify the reagents supplier.

Line 68 - Why was the molar ratio fixed at 0.07? Is there a reference for this methodology?

Line 78 - Avoid using the first person in scientific text.

Line 80 – “Table 25…”. The text seems to be in the wrong place. The methodology for determining water absorption is not well described. Were the samples submitted to what ambient conditions? Were the conditions controlled? Is it possible to repeat the experiment?

Line 97 – Include references for the analytical method

Results and Discussion.

Figure 2 should be in this section. How to explain the order of water absorption of films containing different types of salts? Was it expected? Correlate with the salt chemical characteristics.

Line 109 – Correct: Figure 3 instead of 2.

Line 111 – Is the Figure S1 in the Supp. Material. Clarify for future readers. Remove first person.

Figure 3 – The caption should be self-explanatory. Improve: what are the blue lines?

Line 152 – Why the anion radius or ovality would influence the studied parameters? are there studies that demonstrate this correlation? What salt property would explain this behavior? Charges, basicity and acidity?

References:

5 out of 15 references cited in the manuscript are self-citations. 

Author Response

Dear Reviewer 2

Thank you for your comments. I send you a reply.

Thank you for your kind treatments. I have apologized my delay response. I received the review result on 20th September because of unexpected system errors.
